# Thermal Conductivity of Helium and Argon at High Pressure and High Temperature

**DOI:** 10.3390/ma15196681

**Published:** 2022-09-26

**Authors:** Wen-Pin Hsieh, Yi-Chi Tsao, Chun-Hung Lin

**Affiliations:** 1Institute of Earth Sciences, Academia Sinica, Nankang, Taipei 11529, Taiwan; 2Department of Geosciences, National Taiwan University, Taipei 10617, Taiwan

**Keywords:** high pressure, thermal conductivity, helium, argon

## Abstract

Helium (He) and argon (Ar) are important rare gases and pressure media used in diamond-anvil cell (DAC) experiments. Their thermal conductivity at high pressure–temperature (*P-T*) conditions is a crucial parameter for modeling heat conduction and temperature distribution within a DAC. Here we report the thermal conductivity of He and Ar over a wide range of high *P-T* conditions using ultrafast time-domain thermoreflectance coupled with an externally heated DAC. We find that at room temperature the thermal conductivity of liquid and solid He shows a pressure dependence of *P*^0.86^ and *P*^0.72^, respectively; upon heating the liquid, He at 10.2 GPa follows a *T*^0.45^ dependence. By contrast, the thermal conductivity of solid Ar at room temperature has a pressure dependence of *P*^1.25^, while a *T*^−1.37^ dependence is observed for solid Ar at 19 GPa. Our results not only provide crucial bases for further investigation into the physical mechanisms of heat transport in He and Ar under extremes, but also substantially improve the accuracy of modeling the temperature profile within a DAC loaded with He or Ar. The *P-T* dependences of the thermal conductivity of He are important to better model and constrain the structural and thermal evolution of gas giant planets containing He.

## 1. Introduction

Helium (He), behind hydrogen, is the second most-abundant element in our Universe. Physical properties of ^4^He (the isotope studied in this work) under extreme pressure (*P*) and temperature (*T*) conditions are of great interest in order to understand not only the fundamental physics of such simple molecular systems, but also the structure and evolution of gas giant planets, such as Jupiter and Saturn, that could contain ~20–25 wt.% He [1]. As quantum matter, ^4^He crystalizes in the hexagonal-close-packed (hcp) structure at low and moderate *P-T* conditions (e.g., ~11.6 GPa at room temperature), except for a narrow *P-T* range near its melting curve wherein the body-centered-cubic or face-centered-cubic structure is present [2,3]. In addition to its novel properties [1], ^4^He has also been used as a pressure medium in high *P-T* diamond-anvil cell (DAC) experiments, as it provides nearly hydrostatic conditions within the sample chamber [4]. In past decades, a number of physical properties of ^4^He under various *P-T* conditions have been experimentally and theoretically explored, including the equation of the state and phase diagram [2,3,5,6], melting curve [6,7,8], sound velocity and elastic constants [9,10,11,12,13], refractive index [14], and Raman spectrum [15], etc. Prior studies on the thermal conductivity of He, however, have been limited to ambient or low pressures at a variety of temperature conditions [16,17,18]. A precise determination of He’s thermal conductivity at high *P-T* conditions is critically needed, because it offers an excellent platform from which to explore the physics of thermal transport and phonon interactions in such quantum systems. In addition, it also enables better modeling of He’s thermal conductivity under relevant *P-T* conditions within gas giant planets, advancing our understanding of their thermal and structural evolution dynamics. Furthermore, in high *P-T* DAC experiments loaded with He as a pressure medium, the thermal conductivity of He at a certain *P-T* condition is a key parameter to simulate the heat transfer dynamics and temperature distribution within the DAC. Similar to other pressure media [19,20], such a thermal transport property is crucial for understanding various properties of interest in DAC experiments, such as the thermal conductivity of the sample to be studied.

On the other hand, Ar is also a rare gas at ambient conditions and frequently used as a pressure medium in high *P-T* DAC experiments. Due to its relatively simple interatomic potential, Ar has served as an example material system being used to test the validity of theoretical models. Therefore, various physical properties of Ar under extreme conditions have been extensively studied, see, e.g., [8,21,22,23,24,25,26,27,28,29,30]. In contrast to the largely unknown thermal conductivity of He at high *P-T* conditions, the thermal conductivity of Ar at variable *P-T* conditions has been investigated experimentally and computationally [24,25,29,30]. The thermal conductivity of Ar over a wide range of *P-T* conditions was consistently determined by Green-Kubo molecular dynamics simulations [24] and first-principles calculations [30]. Though optical pump–probe measurements within a DAC [25] showed similar *P-T* dependences to those obtained by the aforementioned computational results, the measured thermal conductivity values are higher. Such a discrepancy may arise from the experimental method or from the interatomic potential or formalism used in the calculations. As a result, a separate, accurate determination of Ar’s thermal conductivity at relevant *P-T* conditions is required to decipher the inconsistency between previous experimental and computational results.

In this paper, we report the thermal conductivity of He and Ar at high *P-T* conditions, such as up to ~50–55 GPa at 300 K, as well as at ~10–19 GPa up to 973 K. These measurements over a wide range of *P-T* conditions enable us to track how the thermal conductivity of He and Ar changes, respectively, upon compression and heating. Interestingly, the thermal conductivity of liquid and solid He scales with a pressure dependence of *P*^0.86^ and *P*^0.72^, respectively, while that of solid Ar scales with a larger pressure slope, *P*^1.25^. Over the temperature range we studied (300–973 K), the thermal conductivity of liquid He at 10.2 GPa increases with temperature via a *T*^0.45^ dependence, whereas that of solid Ar at 19 GPa decreases with temperature via *T*^−1.37^. Our data for solid Ar are in line with previous theoretical calculations. The present results for the *P-T* dependences of the thermal conductivity of He and Ar are significantly important and benefit a variety of studies in materials physics under extreme conditions as well as planetary sciences.

## 2. Methods

### 2.1. Sample Preparation

For high-pressure, room-temperature thermal conductivity measurements, we first polished a thin sheet of borosilicate glass (D 263^®^ T eco from Schott AG, Mainz, Germany), serving as a reference substrate, to ~10 μm thick. We then thermally evaporated ~90 nm-thick Al film as the thermal transducer on the glass substrate, and loaded the glass into a symmetric DAC (300 μm culet) equipped with a Re gasket. The glass substrate was compressed by loading with high-pressure gas of He or Ar (both with a purity of 99.9999%) which serves as the material of interest as well as the pressure medium. To characterize the pressure in the sample chamber of the DAC, we also placed a few ruby balls next to the glass substrate and monitored their pressure-shifted fluorescence or Raman spectrum [31]. The uncertainty of the pressure measurements at room temperature was typically < 5%, depending on the pressure range and medium. As the pressure approached~40–50 GPa, we also compared the pressures determined by the Raman spectra of the ruby and diamond anvil [32], where the difference is typically < 2–3 GPa, depending on the pressure medium. The experimental setup and sample geometry are essentially the same as those in previous studies, see, e.g., [19,20,33].

To create simultaneous high *P-T* conditions, we used an externally heated DAC (EHDAC) [19,34] along with a gas membrane that in situ controls the pressure within the EHDAC during heating. The combination of the EHDAC and gas membrane enabled us to minimize the effects of thermal pressure and study the temperature dependence of the thermal conductivity of Ar and He at a fixed pressure of 19 GPa and 10.2 GPa, respectively, up to ~973 K. Details of the EHDAC assemblage and temperature measurement were described in [19,34].

### 2.2. Thermal Conductivity Measurements

We employed time-domain thermoreflectance (TDTR) to measure the thermal conductivity of He and Ar at high pressures and variable temperatures. In past decades, TDTR has enabled high-precision thermal conductivity measurements of various materials under a wide range of pressure and temperature conditions, with critical applications to physics, chemistry, materials science, and geosciences [35,36,37,38,39]. It is an ultrafast optical pump–probe method, where the output of a Ti:sapphire laser with a central wavelength of 785 nm and a repetition rate of 80 MHz was split into pump and probe beams. In our setup, the pump beam modulated at 8.7 MHz using an electro-optical modulator first passed through a moving mechanical stage that changed its optical beam path. The pump beam was then focused on the Al film coated on the glass substrate, causing variations in Al’s temperature and optical reflectivity. To monitor the heat diffusion dynamics from the Al into the pressure medium and substrate, we focused the probe beam on the Al film and then measured the variation of its reflected intensity, including the in-phase *V_in_* and out-of-phase *V_out_* components, using a fast Si photodiode followed by a lock-in amplifier. More details of the operation principle and setup of TDTR can be found in [40,41].

TDTR data analysis, thermal model fitting, and determination of the thermal conductivity of the sample of interest have been detailed in literature, see, e.g., [19,20,33,37,38] and references therein. For reference, we show a set of representative TDTR data for solid He at 36.1 GPa and room temperature as well as thermal model fittings in Appendix A. Note that in our data analysis, the volumetric heat capacity of each material layer (glass, Al, and He or Ar) within the DAC is an important parameter in the thermal model. Since the volumetric heat capacity of liquid He at high pressure and temperature conditions is not known, it was assumed to be a constant of 4 J cm^−3^ K^−1^, typical of a liquid at high pressure, e.g., [42]. The volumetric heat capacity of solid He at high pressure and room temperature was taken from [9], and its temperature dependence is estimated following a method described in [33]. Compared to the He and Ar, the relatively low thermal conductivity of borosilicate glass that we used as a reference substrate significantly reduces the uncertainty when deriving the thermal conductivity of He and Ar. It is noteworthy that the major uncertainty in our data is from the analysis uncertainty, not the measurement uncertainty. We estimated that the uncertainties in all the parameters used in our thermal model would propagate ~13% error in the derived thermal conductivity of He and Ar before 30 GPa, and ~20% error at 30–50 GPa. An example of detailed TDTR sensitivity analysis and uncertainty evaluations are shown in Appendix A and References [43,44].

## 3. Results and Discussions

Figure 1a presents the pressure dependence of the thermal conductivity Λ(*P*) of He at room temperature. Our measurements started from the liquid phase at 1.3 GPa where Λ = 0.4 W m^−1^ K^−1^. Upon compression, the Λ of the liquid phase increases with pressure to 2.5 W m^−1^ K^−1^ at 11.4 GPa. A pressure-induced solidification of He at ~12 GPa leads to an ~30% increase in the thermal conductivity. The phase of the He (liquid or solid) was in situ identified by time-domain stimulated Brillouin scattering [19] (Appendix A). Similar to the liquid phase, the Λ of polycrystalline, hexagonal-close-packed (hcp) phase increases monotonically with pressure and reaches 8.8 W m^−1^ K^−1^ at 55.2 GPa. Note that compared to the liquid phase, the Λ of the hcp phase shows a slight scattering among different runs of measurement. Such behavior may be associated with the potential anisotropic thermal conductivity of the hcp phase along different orientations of the polycrystalline sample. If we assume that the thermal conductivity of He can be phenomenologically modelled as Λ_He_(*P*) = α_He_*P^n^*, where α_He_ is a normalization constant, then by fitting all the data for each phase in a lnΛ–ln*P* plot, the solid hcp phase has a slightly smaller pressure slope [*n_S_* = 0.72(±0.045)] than the liquid phase [*n_L_* = 0.86(±0.048)], potentially due to the smaller compressibility and anharmonicity in the solid phase.

To understand how the thermal conductivity of He changes with temperature, we have also performed simultaneous high *P-T* measurements (10.2 GPa and 300–973 K) on the liquid phase, see Figure 1b. (We did not perform high *P-T* measurements on the solid phase due to its relatively low melting line [7], where the limited heating range for the solid phase makes a precise determination of the temperature dependence difficult.) Interestingly, the Λ of the liquid phase increases with temperature, consistent with the results at ambient and lower pressures [17,18]. Such behavior, however, is in contrast to the temperature dependence of melts and glasses at higher temperatures [45] and crystalline dielectrics (see, e.g., [46] and Ar below), where their Λ typically decreases as temperature increases. Assuming the thermal conductivity of liquid He scales with *T^m^**^L^*, the exponent value *m_L_* is then determined to be 0.45(±0.036) by a linear slope in its lnΛ-ln*T* plot. Combined with previous temperature-dependence measurements at ambient and lower pressures [17,18], the temperature slope is found to decrease as the pressure increases.

The thermal conductivity of the Ar, on the other hand, shows distinct pressure and temperature dependences compared to the He. Figure 2a presents the Λ(*P*) of Ar at room temperature up to 49 GPa (solid circles). Here, we focus on the solid phase where its Λ = 0.6 W m^−1^ K^−1^ at 2.1 GPa, typical of the relatively low thermal conductivity of molecular solids at similar pressures, see, e.g., CO_2_ [47] and CH_4_ [48]. The Λ increases rapidly with pressure to 30 W m^−1^ K^−1^ at 49 GPa, with a pressure slope of *α* = 1.25(±0.025) in the lnΛ–ln*P* plot. Our data for the values of Λ(*P*) at room temperature and the pressure slope (1.25) are both in good agreement with previous theoretical calculations based on Green-Kubo molecular dynamics simulations (*α* = 1.29) [24] and the Boltzmann transport equation (*α* = 1.39) [30]. Nevertheless, our values for Λ(*P*) are ~50–60% smaller over the pressure range we studied than those calculated by kinetic theory [24] and measured by a similar optical pump–probe technique [25]. In addition, we have further performed high *P-T* measurements to obtain the Λ(*T*) of Ar, see Figure 2b. Two sets of data at 15.8 GPa and 19 GPa up to 973 K consistently show a temperature slope of *β* = −1.26(±0.01) and −1.37(±0.005) in the lnΛ–ln*T* plot, well in line with the temperature dependence reported in literature (*β* ranges from −1.31 to −1.4) [24,25,30]. The deviation of such temperature dependences from the typical *T*
^−1^ has been experimentally observed in solid Ar at ambient [29] and high pressures [25]; such behavior was proposed to be associated with the effects of thermal expansion on the phonon vibrational frequencies [29]. Our results also suggest that the higher pressure may slightly enhance such deviation (from −1.26 to −1.37 when pressure increases from 15.8 GPa to 19 GPa), as discussed in [25].

Note that the opposite temperature dependence of thermal conductivity for liquid He and crystalline Ar originates from different physical mechanisms. The positive temperature dependence for the liquid He is due to the fact that, in liquids, the thermal energy is typically transported by the diffusion of heat from atom to atom, based on the so-called minimum thermal conductivity model, see [49] and references therein. With higher temperature, the stronger vibration of atoms enhances the heat capacity and efficiency of energy transfer between atoms, thus enhancing the thermal conductivity. Moreover, the observation that the pressure decreases the temperature slope for liquid He may be correlated with the increasing fraction of wave-like vibrations (propagons, analogue of the phonons in crystals [49]) that start to contribute to the heat transfer upon compression. The high pressure not only increases the atomic density of the liquid, but also enhances the interatomic interactions. Thus, under compression, the heat transfer mechanism could have transitioned from being primarily the local vibration of atoms, to the combination of it with wave-like vibrations that could propagate through the material. As such, the higher pressure would enhance the scattering of the propagating wave-like vibrations, and therefore suppress the increase in thermal conductivity with temperature (smaller temperature slope). On the other hand, the negative temperature dependence of crystalline Ar is a typical behavior for crystalline materials, where the heat is primarily transported through the phonon scatterings. When the temperature increases, the scattering effects are enhanced, and thus the thermal conductivity is reduced.

A number of phenomenological models have been proposed to describe how the thermal conductivity of a material changes upon compression at variable temperatures. For instance, under the Debye approximation [50,51], thermal conductivity can be expressed as a function of density *ρ* and temperature *T*:(1)$$\Lambda \left(\rho ,T\right)=\Lambda_{0}{\left(\frac{\rho }{\rho_{0}}\right)}^{g}{\left(\frac{T_{0}}{T}\right)}^{n},\;$$
where the $\Lambda_{0}$, $\rho_{0}$, and $T_{0}$ are the reference thermal conductivity, density, and temperature, respectively. Note that the exponent *g* = 3*γ* + 2*q* − 1/3 [50,51], where the Grüneisen parameter γ = (∂lnν/∂lnρ)*_T_*, ν is the phonon vibrational frequency, and *q* = −(*∂*ln*γ*/*∂*ln*ρ*)*_T_*. Based on our experimental results, *n*= −0.45 and about 1.3 for liquid He and solid Ar, respectively, (Figure 1b and Figure 2b). As shown in Figure 3, we derive the *g* values for liquid He, solid He, and solid Ar by plotting their ln(Λ/Λ_0_) − ln(*ρ*/*ρ*_0_) relations, where the $\Lambda_{0}$ and $\rho_{0}$ are the thermal conductivity (Figure 1a and Figure 2a) and density (from [9,23,52]) at 1.3 GPa, 12.9 GPa, and 2.1 GPa for liquid He, solid He, and solid Ar, respectively. As expected, due to the more harmonic interatomic potentials, the *g* = 2.33(±0.15) for the solid He, slightly smaller than that of the liquid He, *g* = 2.57(±0.15). By contrast, the solid Ar has a *g* value of 5.49(±0.2), which is in reasonable agreement with that obtained by molecular dynamics calculations [24] and comparable to that of typical dielectrics, such as MgO [51] and NaCl [19].

Alternatively, the effects of pressure and temperature on the thermal conductivity Λ of a dielectric is often described by the Leibfried-Schlömann (LS) equation [46,53]:(2)$$\Lambda =A\frac{V^{1/3}\omega_{D}{}^{3}}{\gamma^{2}T},$$
where *A* is a normalization constant, *V* the volume, $\omega_{D}$ the Debye frequency, γ the Grüneisen parameter, and *T* the temperature. The LS equation is a simple physical model originally formulated for pure, isotropic dielectric crystals. It assumes that heat is predominantly transported by acoustic phonons via the anharmonic three phonon scattering mechanism. Prior studies have shown that the LS equation adequately describes the pressure dependence of Λ for H_2_O ice VII [42], MgO [51], solid Ar [25], and NaCl in both the B1 and B2 phase [19], at room temperature and over a large range of volume compression (~20–35%). We here also test the validity of the LS equation on the Λ(*P*) of solid He and solid Ar (from 2.1 GPa to 49 GPa with a volume compression of ~50%, larger than that in [25]). As described in [19,42], for simplicity, we assume that the Grüneisen parameter, Poisson ratio, and elastic anisotropy parameter are nearly pressure independent [11,23]. As such, the Λ(*P*) can be simplified to $\Lambda (P)=A^{\prime }V^{5/6}K_{T}{}^{3/2}$ at a constant temperature condition, where $K_{T}$ is the isothermal bulk modulus and $A^{\prime }$ is a normalization constant that will be determined by fitting the present thermal conductivity data at 12.9 GPa and 2.1 GPa for solid He and solid Ar, respectively. Using the equation of state and elastic constants of solid He [9] and solid Ar [23], we plot the Λ(*P*) predicted by the LS equation at room temperature for solid He as the dashed line in Figure 1a, and for solid Ar as the dashed line in Figure 2a. In both cases, the LS equation predicts a larger pressure slope, and its overprediction increases with higher pressure. For instance, the Λ of solid He predicted by the LS equation is ~27%, ~42%, and ~78% larger than our data at 16.1 GPa, 20.5 GPa, and 50 GPa, respectively. Moreover, the predicted Λ for solid Ar based on the LS equation is ~4%, ~35%, and ~72% larger than our data at 5 GPa, 11 GPa, and 38 GPa, respectively. Such differences for solid Ar between the LS prediction and our data at 11–49 GPa are much larger than the uncertainties in our data. Since the Grüneisen parameter γ typically decreases with higher pressure [9,24], the overprediction is expected to be further enlarged. The overprediction of the solid Ar’s thermal conductivity by the LS equation we find here is at odds with the results reported in [25], where the LS prediction roughly captured the trend of their experimental data at 10–50 GPa. The overprediction of the LS equation for the Λ(*P*) of solid He and solid Ar suggests that besides the three phonon anharmonic scattering between acoustic modes, as pressure increases, additional scattering mechanisms that suppress the phonon transport may play an influential role in heat conduction in these materials. Future, more sophisticated computational studies are required to identify their detailed physics of phonon transport.

## 4. Conclusions

We have coupled the ultrafast TDTR and EHDAC to precisely determine the thermal conductivity of He and Ar at high *P-T* conditions. The effects of the density (pressure) and temperature on their thermal conductivity not only shed light on the physical mechanisms of heat transport in these materials, but also enable modelling of their thermal conductivity at further extreme conditions. Our results for liquid He offer critical information to better simulate the thermal conductivity of gas giant planets over a depth range where liquid He is present, such as Jupiter and Saturn which could contain ~20–25 wt.% He and ~75–80 wt.% H_2_ in their interiors. Future studies on the high *P-T* thermal conductivity of H_2_ will significantly improve our understanding of the fundamental physics of heat transport in such simple, quantum systems as well as the thermal evolution of the aforementioned gas giant planets. In addition to He and Ar, knowledge about the thermal conductivity of Ne will also be required. A comprehensive documentation of the thermal conductivity of these important, common pressure media used in high *P-T* DAC experiments will substantially advance research in Earth and planetary sciences, as well as materials physics and chemistry under extreme conditions.

## Figures and Tables

**Figure 1 materials-15-06681-f001:**
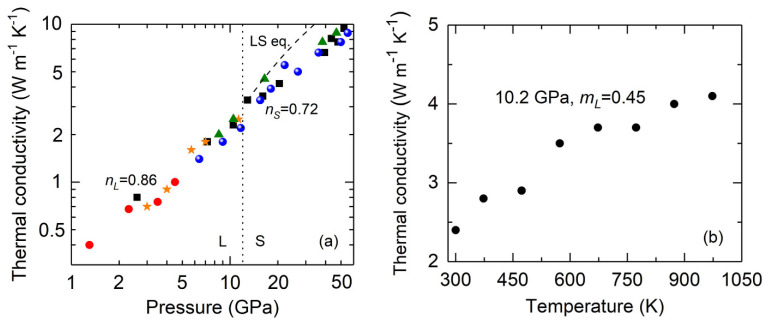
**(a) Thermal conductivity of He at high pressure and room temperature**. Several runs of measurements (represented by different symbols) show consistent results. In both liquid (L) and solid (S) phase, the thermal conductivity increases rapidly with pressure, with a slight increase of ~30% across the liquid–solid transition at ~12 GPa (vertical dotted line). The increasing rate of thermal conductivity with pressure in the solid phase [*n_S_* = 0.72(±0.045)] is slightly smaller than that in the liquid phase [*n_L_* = 0.86 (±0.048)]. The uncertainty is typically ~13% before 30 GPa and ~20% at 30–50 GPa. Compared to our experimental data, the LS equation (dashed curve) predicts a higher thermal conductivity of solid He with a larger pressure slope. **(b) Temperature dependence of the thermal conductivity of liquid He at 10.2 GPa.** The thermal conductivity of the liquid phase (with an analysis uncertainty of ~15%) scales with *T^m^**^L^*, where *m_L_* = 0.45(±0.036).

**Figure 2 materials-15-06681-f002:**
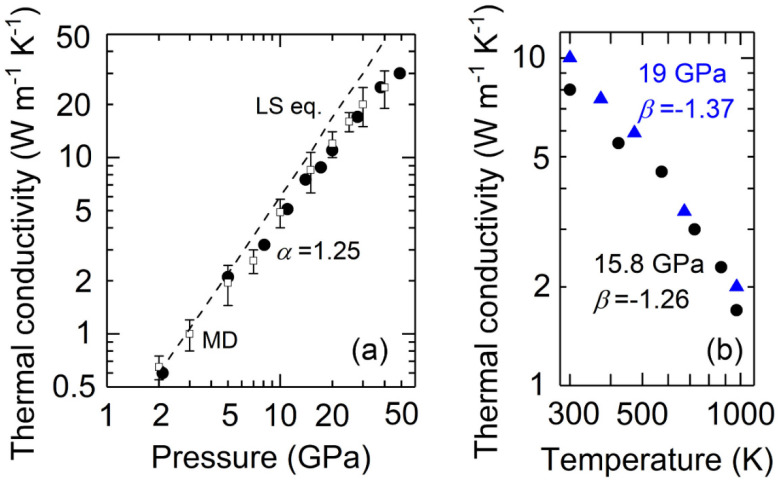
Thermal conductivity of Ar at (**a**) high pressure, room temperature conditions and (**b**) high pressure–temperature conditions. At room temperature, the thermal conductivity (solid circles) increases with pressure with a slope *α* = 1.25 (±0.025) in the lnΛ–ln*P* plot. The open squares and dashed line in (**a**) are results from Green-Kubo molecular dynamics (MD) simulations [24] and the LS equation. Upon heating, the thermal conductivity decreases with temperature following a dependence of *T^β^*, where *β* = −1.26 (±0.01) and −1.37 (±0.005) at 15.8 GPa (black circles) and 19 GPa (blue triangles), respectively.

**Figure 3 materials-15-06681-f003:**
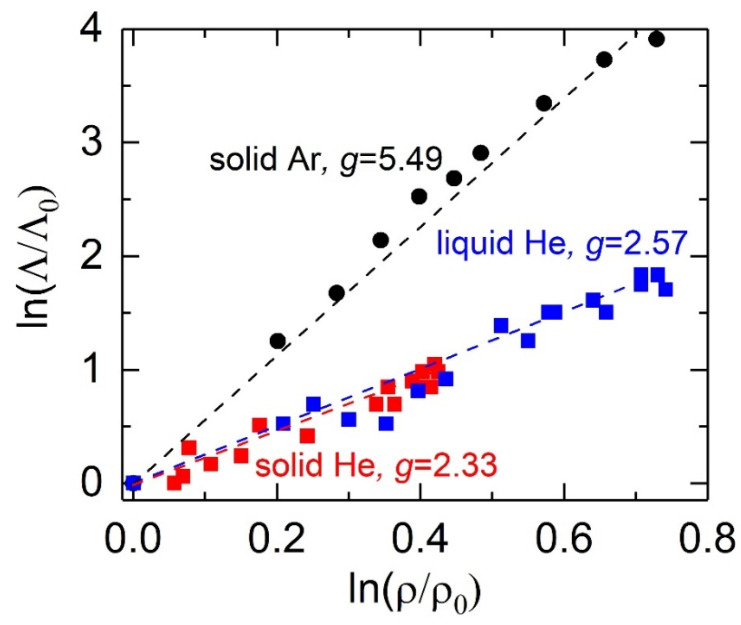
Logarithmic normalized thermal conductivity as a function of logarithmic normalized density for liquid He, solid He, and solid Ar. The blue, red, and black symbols are the present results for liquid He, solid He, and solid Ar, respectively. The dashed line for each set of data is a linear fit with a slope (*g*) of 2.57 (±0.15) (liquid He), 2.33 (±0.15) (solid He), and 5.49 (±0.2) (solid Ar).

## Data Availability

The data that support the findings of this study are available from the corresponding author upon reasonable request.

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
