# Peer review of "Thermal Conductivity of Helium and Argon at High Pressure and High Temperature"

_materials, 2022, doi:10.3390/ma15196681_

Round 1

Reviewer 1 Report

The authors reported thermal conductivity measurements of solid and liquid phases of helium and argon up to ~50 GPa at 300 K and up to ~1000 K at ~10-20 GPa. The authors found the pressure and temperature dependencies of the thermal conductivity of those phases, which can be a benefit for considering the possible temperature distribution in a diamond anvil cell with He and Ar pressure media and for understanding the thermal dynamics in gas planets containing a large amount of both elements. This paper is generally well organized and includes enough data set to support the conclusions of the authors. I recommend it be published in Minerals, but after fixing some of the issues shown below.

On the pressure effect on the thermal conductivity of He, the authors claim that the conductivity of the liquid is more sensitive to pressure relative to the solid phase (L150-). However, I am concerned about whether or not the present results validate the difference. First, the procedure for determining the slopes in Fig. 1(a) is unclear. Were all data points used for the fitting? If so, the slope above 12 GPa looks slightly steeper for me. Or, it seems no differences between solid and liquid when considering the scattering data set, i.e., the errors of the fitted slopes can be too large relative to the possible difference. In addition, I can see some steps at ~12 GPa in green, black and blue symbols, but a step can be recognized at lower pressures in the orange at ~5 GPa and red symbols at ~3 GPa. As the authors explained (L147-), scattering data in the solid phase can be due to a possible preferred orientation of the solid crystals. However, this is not the case with liquids. How do the authors consider the scattering or stepped data points for the liquid phase? Could you please clarify how to obtain the values of the slopes and show the fitted curves in the figure and the errors of the slopes to justify the statement?

Related to the above points, how did the authors confirm the present samples to be liquid or solid? Please represent the phase transition criteria adopted in this work with reasonable data. The statement of the sample is critical for the present investigation.

The authors described the findings of temperature dependency of the thermal conductivity of liquid He (L158-161). However, the positive temperature effect of liquid He conductivity has been reported already, e.g., Blasis and Mann (1960JCP), Arp and McCarthy (1989NIST data book) cited in this paper. These previously reported data at 1 bar or relatively low pressures to the present 10 GPa data. Therefore, I recommend the authors discuss how the pressure, in other words, density, affects the positive temperature dependency of the conductivity. Furthermore, the authors should discuss the positive temperature effect of liquid He and the negative of Ar from the Physical viewpoint such as the molecular structure, atomic dynamics, and the mechanics contributing to each thermal conductivity.

By the way, the thermal conductivities at high temperatures have been confirmed to be reversible upon heating and cooling? Upon heating with EH-DAC, a deformation of the sample chamber or chemical reaction could be possible, which might affect the conductivity of the liquid sample. The reversibility of the heating and cooling cycles can rule out or decrease the risk by changing sample conditions upon heating.  

Other points:

The manuscript's statistical uncertainties for some fitted or obtained parameters are missing. Please carefully check them.

L215: This means the conductivity of liquid He is more sensitive to compression than solid. It sounds opposite to comparing the slopes in the conductivity over pressure plots.

L244: How about a possible impact on the differences between the LS mode and the present data by assuming the used parameters such as Grüneisen parameters, Poisson ratio, and elastic parameters?

Table S1: How did the authors determine the sample's and glass's thickness during compression and/or heating? The uncertainties of the thickness estimation may be little effect on the reported conductivities. Please describe the procedure in the text or supplement. I believe it is not easy to measure the thickness of the samples in DAC at high-P-T conditions.

Reviewer 2 Report

The authors discuss helium and argon's thermal conductivity at different high pressures and temperatures. I wish to appreciate this study as the study focused on the effect of pressure on thermal conductivity of, Helium (He) and Argon (Ar). Observation and suggestions are as follows

·       in line 69: The author has mentioned that he reported the results for ~50-55 GPa at 300K. But in the result discussion, the thermal conductivity for 55 GPa is missing.

·       The experimental test rig may be shown with labels for better understanding to readers.

·       Line 134-136: "We estimated that the uncertainties in all the parameters used in our thermal model would propagate ~13% error in the derived thermal conductivity of He and Ar before 30 GPa, and ~20% error at 30–50 GPa". What is your justification for varying such high uncertainty 13-20% measurement errors?

·       In the result and discussion section, the author tries to justify most of the findings qualitatively with previous studies. Comparing the results quantitatively with suitable previous results or findings is suggested.
